# Few-Shot Hyperspectral Image Classification Based on Convolutional Residuals and SAM Siamese Networks

**Mengen Xia [1], Guowu Yuan [1], Lingyu Yang [1], Kunming Xia [1], Ying Ren [2], Zhiliang Shi [2] and Hao Zhou [1,***

[1] School of Information, Yunnan University, Kunming 650504, China; charmenten@163.com (M.X.); gwyuan@ynu.edu.cn (G.Y.); proyang98@gmail.com (L.Y.); xiakunming@stu.ynu.edu.cn (K.X.)
[2] Kunming Enersun Technology Co., Ltd., Kunming 650504, China; renying@enersun.cn (Y.R.); shizhiliang@enersun.cn (Z.S.)
[*] Correspondence: zhouhao@ynu.edu.cn; Tel.: +86-87165033748

**Abstract:** With the development of few-shot learning, significant progress has been achieved in hyperspectral image classification using related networks, leading to improved classification outcomes. However, practical few-shot hyperspectral image classification encounters challenges such as network overfitting and insufficient feature extraction during the model training process. To address these issues, we propose a model called CRSSNet (Convolutional Residuals and SAM Siamese Networks) for few-shot hyperspectral image classification. In this model, we deepen the network depth and employ the convolutional residual technique to enhance the feature extraction capabilities and alleviate the problem of network gradient degradation. Additionally, we introduce the Spatial Attention Mechanism (SAM) to effectively leverage spatial information features in hyperspectral images. Lastly, metric learning is employed by comparing the distance between two output feature vectors to determine the label category. Experimental results demonstrate that our method achieves superior classification performance compared to other methods.

**Keywords:** hyperspectral image classification; few-shot learning; metric learning; Siamese networks; space attention mechanism

## 1. Introduction

Hyperspectral images (HSI) are obtained and compiled using hyperspectral sensors or imaging spectrometers, which contain numerous continuous bands that offer abundant spectral and spatial information. As a result, hyperspectral images find valuable applications in mineral resources [1], agricultural production [2], environmental monitoring [3,4], and astronomy [5]. Previous studies have employed various methods for hyperspectral image classification, as depicted in Figure 1a, including Support Vector Machines (SVM) [6], Random Forests (RF) [7], sparse representation-based [8] and K Nearest Neighbors (KNN) [9]. However, these classification approaches are not well-suited for multi-classification problems and fail to effectively leverage the high-dimensional hyperspectral data. Consequently, they do not fully exploit the abundant spectral and spatial information available in hyperspectral images, leading to unsatisfactory classification results.

With the growing significance of deep learning in computer vision, deep-learning models exhibit greater network depth and enhanced data-mining capabilities compared to traditional network structures [10]. In the realm of deep learning, various structural models have been applied to hyperspectral image classification, as illustrated in Figure 1b, including Convolutional Neural Networks (CNN) [11], Stacked Autoencoder Networks (SAN) [12], Deep-Belief Networks (DBN) [13], and Recurrent Neural Networks (RNN) [14]. Among them, the Convolutional Neural Network (CNN) has emerged as the primary method for hyperspectral image classification. Yang et al. [15] proposed a multilevel spectral–spatial transformation network (MSTNet) for hyperspectral image classification

(HSIC). The network utilizes transformer encoders to learn feature representations and decoders to integrate multi-level features, resulting in accurate classification results. VG-GNet [16], introduced by Simonyan K et al., utilizes small convolutional kernels ($3 \times 3$) and small pooling kernels ($3 \times 3$), allowing for deeper network models while managing computational growth. ResNet [17], also devised by He K M et al., enhances overall network performance through residual learning using convolutional layers and increased network depth. These advancements aim to improve the effectiveness of hyperspectral image classification. Zhu et al. [18]. introduced a short-range and long-range graph convolution (SLGConv) based on the graph convolutional neural network. They utilized a three-layer SLGConv to construct the short- and long-range graph convolution network (SLGCN) for extracting both global and local spatial–spectral information to improve hyperspectral image classification. However, Liao et al. [19] proposed a spectral–spatial fusion transformer network (S2FTNet) for HSI classification. S2FTNet leverages the transformer framework to create a spatial transformer module (SpaFormer) and a spectrum converter module (SpeFormer) to capture long-distance dependencies between image space and spectrum, enhancing the classification performance.

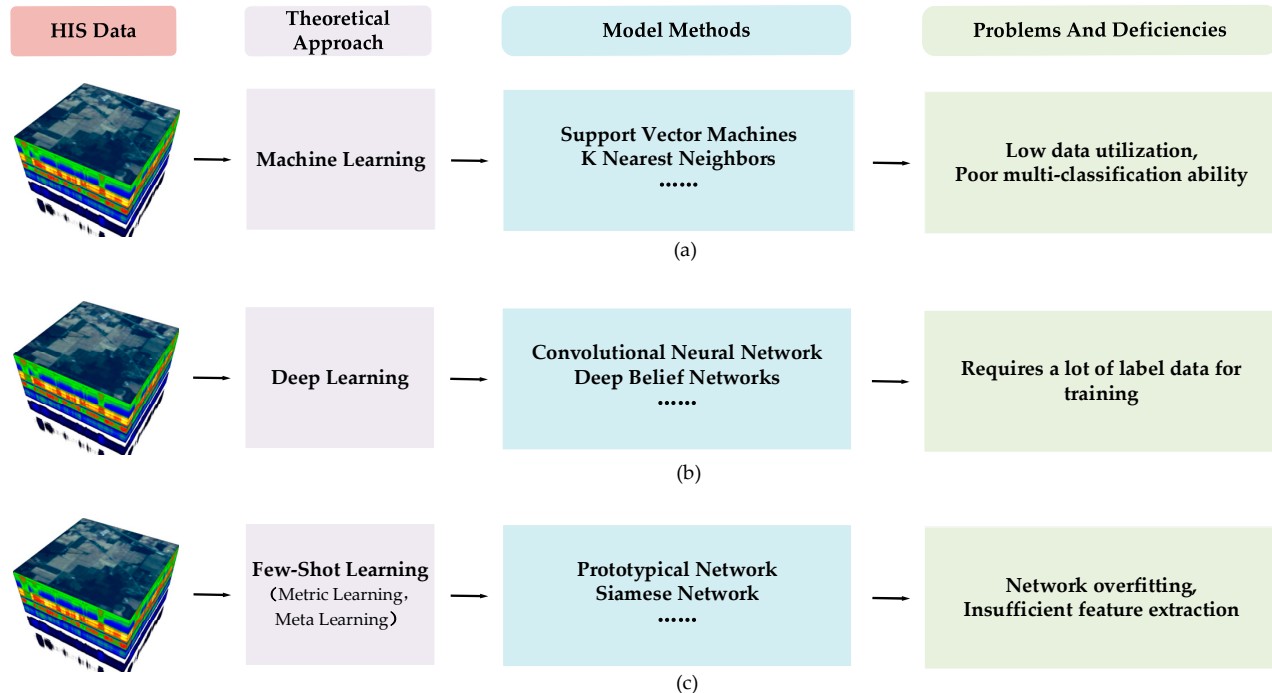

**Figure 1.** The main classification methods of hyperspectral images: (**a**) Hyperspectral image classification based on machine learning; (**b**) Hyperspectral image classification based on deep learning; (**c**) Hyperspectral image classification based on few-shot learning.

Although deep learning demonstrates excellent performance in hyperspectral image classification, its effectiveness heavily relies on a substantial number of labeled samples. Unfortunately, the process of labeling hyperspectral image samples is labor-intensive and costly, resulting in the limited availability of labeled data [20].

The scarcity of labeled samples poses challenges such as overfitting the deep-learning model and reduced accuracy. Consequently, the primary focus of future hyperspectral image classification lies in achieving satisfactory results with limited samples. Finding ways to optimize deep-learning models under such circumstances is a key research direction in the field.

In recent years, few-shot learning has gained popularity and is commonly applied to few-shot target classification tasks. Two prominent approaches in few-shot learning are metric learning and meta learning, as depicted in Figure 1c. Metric-learning methods focus

on learning feature embeddings or similarity measures to enable classification based on discriminative similarities between query set labels and support set-labeled examples. These methods often employ episodic training mechanisms to acquire transferable knowledge. Meanwhile, the meta-learning method aims to learn an optimal initialization or optimizer that can generalize the model to unseen tasks. For instance, recurrent neural networks are employed to train cross-task meta-learners, enhancing knowledge transfers across different tasks and improving generalization performance. Both approaches share the common objective of learning a model capable of classifying query set labels with limited support set labels.

This paper combines various common classification methods for hyperspectral imagery with approaches used to address the issue of limited samples in traditional image processing. The main structure of this article is outlined as follows:

- Related work: This section primarily focuses on presenting the relevant theories of metric learning and identifying the deficiencies in existing partial networks, subsequently leading to the introduction of the main innovations of our proposed network.
- Materials and methods: We provide a concise overview of the designed network architecture, followed by a more detailed exposition of the concepts related to the spectral module, spatial module, and loss function.
- Experimental results: This section encompasses the introduction of three commonly used hyperspectral image datasets, the description of the relevant experimental procedures, and a comprehensive analysis and discussion of the experimental results.
- Conclusions: This section summarizes the main findings of the study, reviews the key points of the paper, and identifies the current limitations of the research, followed by proposing suggestions for future work.

## 2. Related Work

### 2.1. Few-Shot Learning Based on Metric Learning

Metric learning, also known as similarity learning, is a type of the few-shot learning approach based on migration learning [21]. It involves calculating the distance between the sample to be classified and known samples, analyzing the distances to measure their similarity, and finding the nearest matching sample for classification [22].

The Siamese network is a neural network model based on metric learning. In this model, pairs of samples are fed into the dual-channel Siamese network for feature extraction. For the input sample pair belonging to the same class, it is labeled as a positive sample (Label 1), while input sample pairs from different classes are labeled as negative samples (Label 0). The ratio of positive and negative samples has an impact on the network's feature extraction classification accuracy. To ensure a balanced effect, this paper maintains a 1:1 ratio between positive and negative samples. The input data is then transformed into a target space using the Siamese network. Subsequently, a similarity calculation is performed using the distance function. This involves computing the distance between the output feature vectors of the two samples being tested and the feature vectors of the known samples in the feature vector space. The known labeled sample with the smallest distance to the test sample is selected as its classification class, thus completing the classification process.

The utilization of the dual-channel Siamese network, which employs metric learning in the few-shot learning method, leads to enhanced classification performance on samples. In contrast to the traditional single-channel network as depicted in Figure 2, which solely focuses on simple inter-class sample classification, it lacks the ability to effectively discern relationships and distinguish between different samples. However, the dual-channel Siamese network addresses this limitation by establishing stronger connections within the same categories while expanding the separation between different categories. This is accomplished by leveraging input sample pairs that facilitate differentiation of various categories, consequently leading to improved classification outcomes.

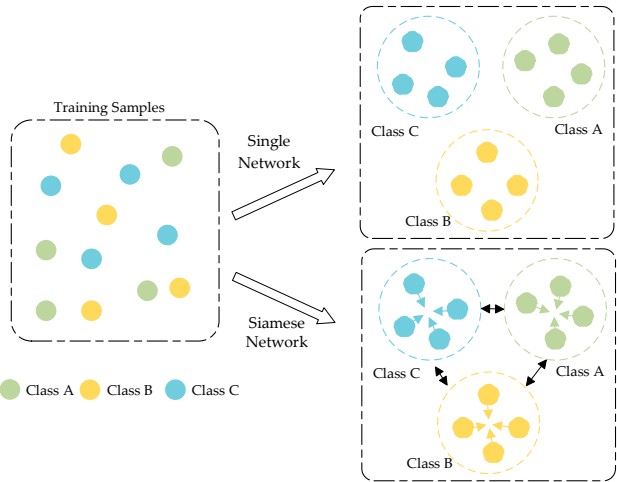

**Figure 2.** Difference between metric learning-based single-channel neural network and dual-channel Siamese network.

## 2.2. Related Network Models

There are three main types of few-shot networks currently applied to hyperspectral image classification [23]: prototypical networks, relation networks, and Siamese networks.

Tang et al. [24]. proposed a novel multiscale spatial spectroscopic PN (MSSPN). The network is a multi-scale spatial spectral feature extraction algorithm based on a trapezoidal structure, effectively achieving fusion of spatial spectral features at different scales. It also designs a multi-scale spatial spectral prototype representation based on the trapezoidal structure extraction algorithm theory, which offers improved scalability and better classification effectiveness. Zhang [25] proposed a "Global Prototypical Network" for hyperspectral image classification using a prototype network. Although the current prototype network shows effectiveness in classifying hyperspectral images, its generalization ability to new samples is limited. When encountering unseen samples, the classification performance trend decreases.

Sung F et al. proposed the relation network [26], in which the network initially processes randomly selected samples from the query set and sample set through the embedding layer to process and obtain the feature map. These feature maps are then concatenated and passed through the relation network to calculate the relation score, determining whether the samples belong to the same category. Gao et al. [27] employed a meta-learning strategy to design a feature-learning module and a relation-learning module, aiming to enhance the effectiveness of hyperspectral classification. However, the method based on the relational network relies on the selection and quantity of sample data. An insufficient sample size, or poor sample quality, can lead to the underutilization of hyperspectral image sample information in small sample scenarios, impacting feature extraction capabilities and resulting in decreased classification accuracy.

Wang et al. [28] propose a Siamese CNN with a soft-loss function that adapts based on coupling substitution data enhancement. This approach effectively addresses the HSI classification challenge with limited training samples. Huang et al. [29] utilized a bidirectional Siamese network to separately fuse spectral space features, thereby enhancing the network's capability for feature extraction in hyperspectral images. However, the Siamese network is susceptible to overfitting or underfitting when dealing with small samples. In the context of hyperspectral image classification, the high dimensionality of the data and the limited number of samples can lead to good performance on the training set but poor performance on the test set.

To address the aforementioned challenges, this paper presents a few-shot hyperspectral image classification based on convolutional residuals and SAM Siamese networks. The key contributions are as follows:

- We propose a novel dual-channel Siamese network architecture for spatial–spectral feature extraction in hyperspectral images. This model leverages convolutional neural networks of different dimensions, employing one-dimensional convolution for spectral feature extraction and two-dimensional convolution for spatial feature extraction.
- We introduce a feature residual-extraction module specifically tailored for hyperspectral image classification. This module not only reduces the network's 0complexity, but it also leverages multi-layer features to effectively extract spectral–spatial information.
- In order to enhance the utilization of spatial information, we incorporate a Spatial Attention Mechanism (SAM) into the spatial feature-extraction module. Additionally, to mitigate overfitting caused by network deepening and improve the model's generalization ability, we adopt a superior label-smoothing cross-entropy loss function [30].

## 3. Materials and Methods

The network architecture of this study is depicted in Figure 3. Initially, the hyperspectral image data samples are divided into a training set and test set. Each data sample is further segmented into data blocks of varying sizes, denoted as W1 × H1 for larger data blocks and W2 × H2 for smaller data blocks. During the training process, the data blocks of all samples are randomly selected and paired to create positive and negative sample pairs. These pairs are then input into the feature extraction network of the dual-channel Siamese network to extract deeper spectral and spatial information of hyperspectral data. The network model is continuously optimized through metric learning, which involves extracting feature vectors from hyperspectral image data. This enables the network to learn the classification ability for different classes. The parameters of the feature extraction module are adjusted via back-propagation of the loss function, facilitating the fine-tuning of the network.

During the testing process, the sample data in the test set is divided into a support set and a query set. The query set consists of the data to be detected, while the support set consists of the known class data. These two sets form a sample pair, where each pair is composed of a data block from the query set and a data block from the support set. The sample pairs are then input into the trained model. The trained model outputs feature vectors for each sample pair. The distance between the feature vector of the data in the query set and the feature vector of the data in the support set is calculated. The sample pair with the highest similarity in terms of feature vectors, specifically between the feature vectors of the support set and the query set, is selected. The predicted class for the data in the query set is determined based on this selected sample pair. This process completes the classification of hyperspectral images.

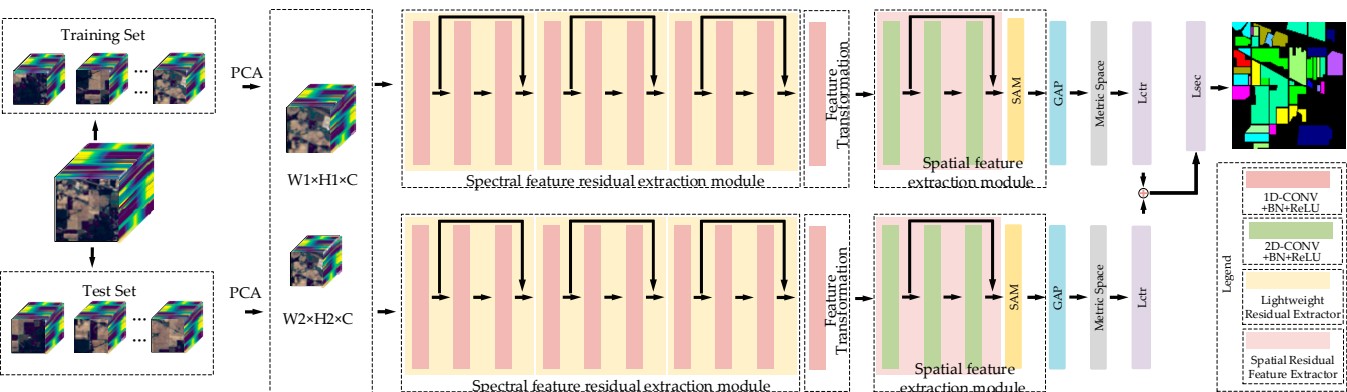

**Figure 3.** CRSSNet network structure.

### *3.1. Model Method*

3.1.1. Spectral Feature Residual Extraction Module

In the scenario of limited samples, the deep-learning network often encounters the issue of vanishing gradients as the number of network layers increases. To address this problem, the convolution residual method is incorporated into the spectral extraction module, which helps mitigate gradient degradation during the deepening of network. A lightweight implementation of the residual method is adopted. Each lightweight residual extractor, as depicted in Figure 4, comprises convolutional layers, Batch Normalization (BN) layers, and ReLU layers, applied in sequence.

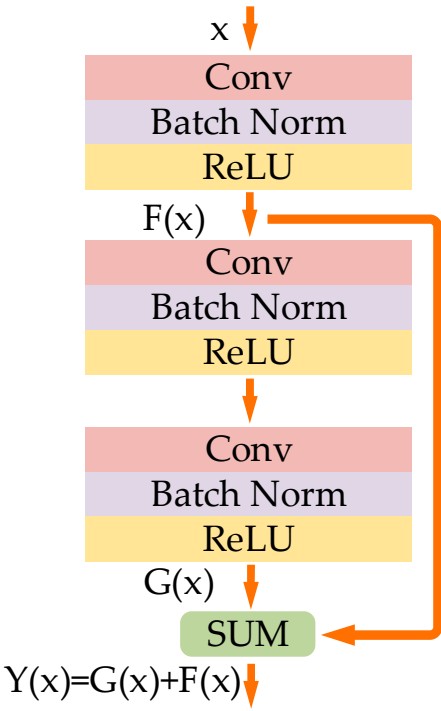

**Figure 4.** Lightweight residual extractor.

As illustrated in Figure 4, the initial data input to the lightweight residual extractor is denoted as $x$. Each convolutional layer is subsequently followed by a Batch Norm layer [31], facilitating network convergence and improving generalization ability. This data processing results in an output following a normal distribution with a mean value of 0 and a variance of 1. Furthermore, a Rectified Linear Unit (ReLU) activation function is employed to reduce network computational complexity and address the issue of gradient vanishing in deep network.

$$Y(x) = G(x) + F(x) \tag{1}$$

The spectral residual-extraction module is depicted in Figure 5. In the first lightweight residual extractor, the convolution has a kernel size of $3 \times 1 \times 1$ and 16 channels. For the second lightweight residual extractor, the convolution layer has a kernel size of $3 \times 1 \times 1$ and 32 channels. The third lightweight residual extractor has a convolution layer with a kernel size of $3 \times 1 \times 1$ and 64 channels. Additionally, the output of the first ReLU activation function $F(x)$ is added to the output of the third ReLU activation function, $G(x)$, to obtain the overall output $Y(x)$. The computation process is shown in Equation (1). The output $Y(x)$ is passed to the next lightweight residual extractor. The feature map obtained after extraction by three lightweight residual extractors is denoted as $Z \in R^{(W \times H \times 1 \times 64)}$.

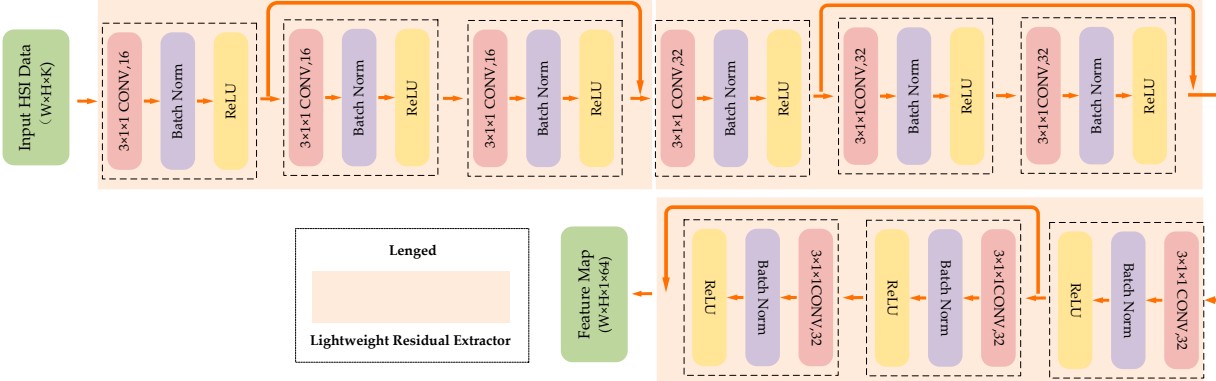

**Figure 5.** Spectral residual-extraction module.

Due to the limited number of samples in the few-shot scenario, it is important to control the number of parameters in the network model. Therefore, in the spectral feature-extraction part of the network, the special size of the three-dimensional convolution is set to 1, which effectively reduces the model parameters by 2/3.

The implementation of one-dimensional convolution in the network involves setting the spatial size of the three-dimensional convolution to 1, as shown in Equation (2). The value $v_{ij}^{xyz}$ of the one-dimensional convolution represents the computation of the neuron at position $(x, y, z)$ in the $j$-th feature map of the $i$-th layer and is provided by the following equation:

$$v_{ij}^{xyz} = f(b_{ij} + \sum_m \sum_{p=1}^{1} \sum_{q=1}^{1} \sum_{r=0}^{R_i-1} w_{ijm}^{pqr} v_{(i-1)m}^{xy(z+r)}) \tag{2}$$

where $m$ represents the index of the feature map connected to the $j$-th feature map in the $(i-1)$-th layer, $R_i$ is the size of the convolution kernel along with the spectrum dimension; $p$ denotes the length of the spatial convolution kernel, $q$ denotes the width of the spatial convolution kernel, and $p$ and $q$ are both set to 1; $w_{ijm}^{pqr}$ is the value connected to the position $(p, q, r)$ in the $m$-th feature map; $b_{ij}$ is the bias of the $j$-th feature map in the $i$-th layer; the function $f(\bullet)$ is the ReLU activation function.

### 3.1.2. Spatial Feature Extraction Module

In scenarios of limited samples, effectively utilizing spatial information can significantly enhance hyperspectral image-classification capabilities. Based on the structural characteristics of HSI, this paper proposes a novel spatial feature extractor, depicted in Figure 6. The spatial feature extractor primarily consists of a spatial residual feature extractor and a spatial attention mechanism.

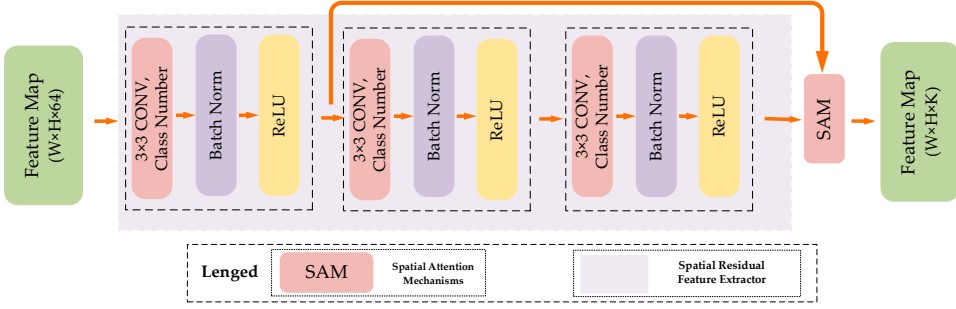

**Figure 6.** Spatial feature extraction module.

Spatial Residual Feature Extractor

The spatial feature-extraction component is illustrated in Figure 6. The spatial feature-extraction module effectively captures the spatial information of the feature map obtained from generating the spectral spatial-connection layer. This is achieved by employing a combination of a two-dimensional convolution residual block and a spatial attention mechanism. The resulting feature map generated by this module contains discriminative information related to different classes of hyperspectral images. In this process, the feature map $Z \in R^{(W \times H \times 1 \times 64)}$ is transformed into $Z \in R^{(W \times H \times K)}$ through the utilization of this module, where $K$ represents the dimension of the feature map.

Using a similar approach to one-dimensional convolution, the spatial feature extraction component sets the spatial size of the three-dimensional convolution to $3 \times 3$, forming a two-dimensional convolution. The $v_{ij}^{xy}$ of the two-dimensional convolution represents the calculated value of the neuron at the position $(x, y)$ in the $j$th feature map of the $i$th layer, as shown in Equation (3):

$$v_{ij}^{xy} = f\left(b_{ij} + \sum_{m} \sum_{p=0}^{p_i-1} \sum_{q=0}^{q_i-1} w_{ijm}^{pq} v_{(i-1)m}^{(x+p)(y+q)}\right) \tag{3}$$

where $m$ represents the index of the feature map connected to the $j$-th feature map in the $(i-1)$-th layer; $p_i$ represents the length of the spatial convolution kernel in the $i$-th layer, and $q_i$ represents the width of the spatial convolution kernel in the $i$-th layer; $w_{ijm}^{pq}$ represents the value connected to the position $(p, q)$ of the $m$-th feature map; $b_{ij}$ is the bias of the $j$-th feature map in the $i$-th layer; and the function $f(\bullet)$ represents the ReLU activation function.

Spatial Attention Mechanisms

The feature map $Z \in R^{(W \times H \times K)}$, obtained from the two-dimensional convolutional layer, serves as the input feature map of the spatial attention mechanism, as depicted in Figure 7 below. Initially, the input feature map is fed into the spatial attention mechanism, where maximum pooling and average pooling operations are performed to capture different information. This result in two feature maps of $Z \in R^{(W \times H \times 1)}$, each of size $W \times H \times 1$, where every pixel in the generated image incorporates features from all channels at that position.

Then, the two generated feature maps $Z$, both of size $Z \in R^{(W \times H \times 1)}$, are then concatenated along the channel dimension. Subsequently, a $7 \times 7$ convolutional layer is applied to transform the feature map into a single-channel representation $Z$ of size $Z \in R^{(W \times H \times 1)}$. The Sigmoid activation function is utilized to map the pixel values in the feature map to the probability space of 0 to 1, capturing the more prominent feature information in the image. This mapping generates spatial attention weight coefficients $M_\text{s}$. Finally, the attention weights are multiplied element-wise, channel-by-channel, with the input feature map of the module, denoted as "$M_\text{s} \times$ input features", resulting in a new feature $Z$ of size $Z \in R^{(W \times H \times K)}$. The attention mechanism is expressed by Equation (4).

$$M_\text{s}(F) = \sigma\left(f^{7 \times 7}([AvgPool(F), MaxPool(F)])\right) \tag{4}$$

where $M_\text{s}$ represents the attention weight coefficient, $F$ represents the input feature, $\sigma$ represents the Sigmoid activation function, $f^{7 \times 7}$ represents the $7 \times 7$ convolution kernel, and AvgPool and MaxPool represent average pooling and maximum pooling respectively.

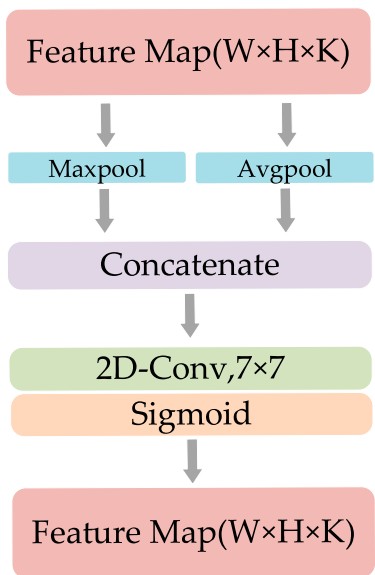

**Figure 7.** Spatial Attention Mechanisms.

### 3.2. Loss Function

In the training process of the dual-channel Siamese network, the loss function is calculated using a weighted-comparison loss function and a label-smoothing cross-entropy loss function for each training iteration. The total loss function is expressed as Equation (5):

$$L = L_{ctr} + L_{sce} \tag{5}$$

where $L_{ctr}$ is the weighted contrast loss function, and $L_{sce}$ is the label-smoothing cross-entropy loss function.

The dual-channel Siamese network outputs two feature maps of different sizes, $Z_1 \in \mathbb{R}^{H_1 \times H_1 \times K}$ and $Z_2 \in \mathbb{R}^{H_2 \times H_2 \times K}$, and extracts their center vectors, $L_1 \in \mathbb{R}^{1 \times K}$ and $L_2 \in \mathbb{R}^{1 \times K}$, respectively, where $H_1$ and $H_2$ represent the sizes of two feature maps. The cosine distance $w$ between two input sample pairs can be calculated using Equation (6), and the weighted comparison loss function is defined by Equation (7):

$$w = \frac{L_1 \times L_2}{\|L_1\| \|L_2\|} \tag{6}$$

$$L_{ctr}(y, d_{pos}, d_{neg}) = y \times (1 - w) \times d_{pos} + (1 - y) \times w \times \max(m - d_{neg}, 0) \tag{7}$$

Equation (7) represents the weighted comparison loss function, where $(1 - w)$ is multiplied by the positive sample $y$ pair, and $w$ is multiplied by the negative sample pair; $d_{pos}$ denotes the distance between the feature vectors of the same class samples in a sample pair, while $d_{neg}$ represents the distance between the feature vectors of different class samples in a sample pair. The variable $m$ represents the maximum distance between different samples.

The dual-channel Siamese network receives different information, resulting in a diversification of the output feature vector. To optimize the network parameters effectively, noise is introduced simultaneously to prevent excessive confidence in the correct label and enhance the network's generalization ability. Hence, the label-smoothing cross-entropy loss function is introduced, defined by Equation (10) as follows:

$$p_k = \frac{e^{x^T w_k}}{\sum\limits_{l=1}^{L} e^{x^T w_l}} \tag{8}$$

$$y_k = y_k(1 - \alpha) + \frac{\alpha}{K} \tag{9}$$

$$L_{sce} = \sum_{k=1}^{K} -y_k \log(p_k) \tag{10}$$

In Equation (8), $p_k$ represents the probability of each category of data, $w_k$ corresponds to the weights associated with the $k$-th category sample pairs, and $x$ is the vector of activations from the second-to-last layer of the network. $T$ denotes the transpose of $x$.

In Equations (9) and (10), $y_k$ represents the label vector, $\alpha$ is the label-smoothing factor, and $K$ represents the total number of categories being classified.

## 4. Experimental Results

### 4.1. Experimental Datasets

To evaluate the efficacy of the proposed few-shot hyperspectral image classification method, three publicly available hyperspectral image datasets were chosen for experimental validation: the University of Pavia, Indian Pines, and Salinas [32]. The University of Pavia and Salinas datasets represent agricultural farmland hyperspectral image datasets, while the University of Pavia dataset pertains to an urban hyperspectral image dataset. Table 1 provides detailed information regarding these three public datasets, including the pseudo-color image and the ground-truth image. Please refer to Figure 8 for visual representations of the datasets.

**Table 1.** Details of the three hyperspectral image datasets.

|  | Indian Pines | Pavia University | Salinas |
|---|---|---|---|
| Pixel resolution | $145 \times 145$ | $610 \times 310$ | $512 \times 217$ |
| Spectral range (nm) | 400–2500 | 430–860 | 400–2500 |
| Number of bands | 200 | 103 | 204 |
| Spatial resolution (m) | 20 | 1.3 | 3.7 |
| Sensor | AVIRIS | ROSIS | AVIRIS |
| Number of categories | 16 | 9 | 16 |
| Total number of pixels | 21,025 | 207,400 | 111,104 |

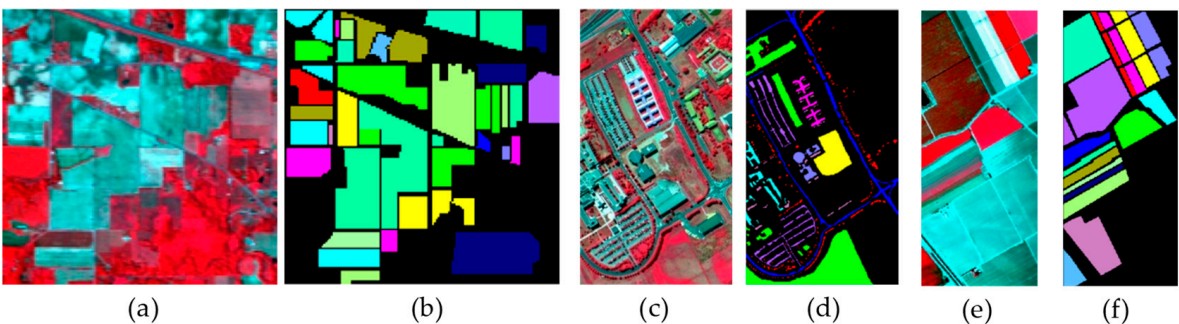

(a)      (b)      (c)      (d)      (e)      (f)

**Figure 8.** Indian Pines datasets. (**a**) False-color composite image; (**b**) Ground-truth map; University of Pavia datasets. (**c**) False-color composite image; (**d**) Ground-truth map; Salinas datasets. (**e**) False-color composite image; (**f**) Ground-truth map.

### 4.2. Experiment Setup

The experimental hardware computer configuration consists of Intel® Core™i7-6700 K CPU @4.00 GHz RAM 32 GB and NVIDIA TITAN X (Pascal) 12 GB graphics card. The training and testing are conducted using the Pytorch neural network framework.

In the network, the Patch Size for hyperspectral input is set to $13 \times 13$ and $7 \times 7$ for two channels [23]. During training, five labeled samples are randomly selected for each class for model training, while the remaining samples are used for testing. The model is

optimized by the Adam optimizer, with a learning rate of 0.001. This setting effectively trains the network, accelerates its convergence, and reduces the training time. To mitigate the impact of random results on the classification performance, all experimental data in this study are averaged over 10 experimental results obtained with randomly selected samples.

In this paper, the quantitative evaluation indicators for assessing the performance of hyperspectral image classification methods are as follow: overall accuracy (OA), average accuracy (AA), and the Kappa coefficient. The overall accuracy represents the ratio of correctly classified pixels to the total number of tested pixels in the hyperspectral data. Its calculation equation for overall accuracy is as follows:

$$OA_i = \frac{C_{ii}}{N} \tag{11}$$

In the Equation, $N$ represents the total sample size, and $C$ represents the confusion matrix of size $n \times n$.

The average classification accuracy represents the average accuracy of each category's classification. Its calculation equation is as follows:

$$AA = \frac{\sum\limits_{i=1}^{n} OA_i}{N} \tag{12}$$

In the Equation, $N$ represents the total number of samples, and $n$ represents the number of hyperspectral image categories.

The Kappa coefficient is used as an indicator to measure the agreement between the classification results of the hyperspectral dataset and the actual effect. Typically, the Kappa coefficient ranges from 0 and 1. A Kappa value of 1 indicates complete agreement, Kappa $\geq 0.75$ indicates satisfactory agreement, and Kappa $< 0.4$ indicates a less-than-ideal agreement. The calculation equation for the Kappa coefficient is as follows:

$$Kappa = \frac{N \sum\limits_{i=1}^{n} C_{ii} - \sum\limits_{i=1}^{n} C_{ii} \times N_i}{N^2 - \sum\limits_{i=1}^{n} C_{ii} \times N_i} \tag{13}$$

In the Equation, $N$ represents the total number of samples, $n$ represents the number of hyperspectral image categories, $N_i$ represents the number of samples of the $i$-th class, and $C$ represents the confusion matrix of size $n \times n$.

To evaluate the impact of label factors in the label-smoothing loss function on OA, AA, and Kappa coefficients, this study conducts experiments on three datasets by varying the label factor from 0.1 to 0.9. The experimental results are depicted in Figure 9. It is observed that when the label factor is set to 0.2, the OA value is generally higher than other parameter settings. Therefore, for subsequent ablation experiments and comparative experiments, a label factor of 0.2 is chosen.

### 4.3. Ablation Experiments

To verify the improvement in overall accuracy of the residual feature extractor and the spatial attention mechanism in the proposed CRSSNet method, it is compared with the enhanced network CRSSNet (no SAM) that does not utilize the spatial attention mechanism, as well as the unimproved original network S3net. Two sets of ablation experiments are conducted on three datasets to validate the effectiveness of the residual feature extractor and spatial attention mechanism in the CRSSNet method. The flat average OA, AA, and Kappa data from 10 experiments of the two ablation models are presented in Tables 2–4, respectively.

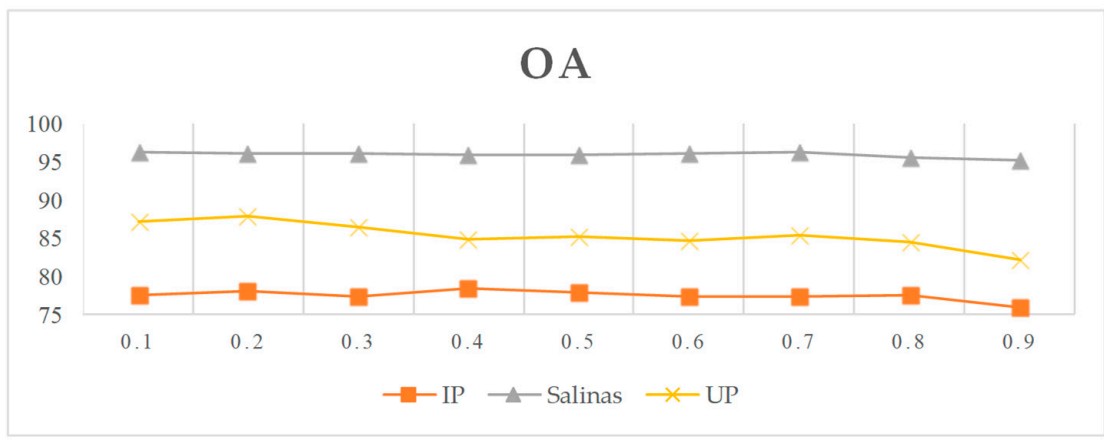

(a)

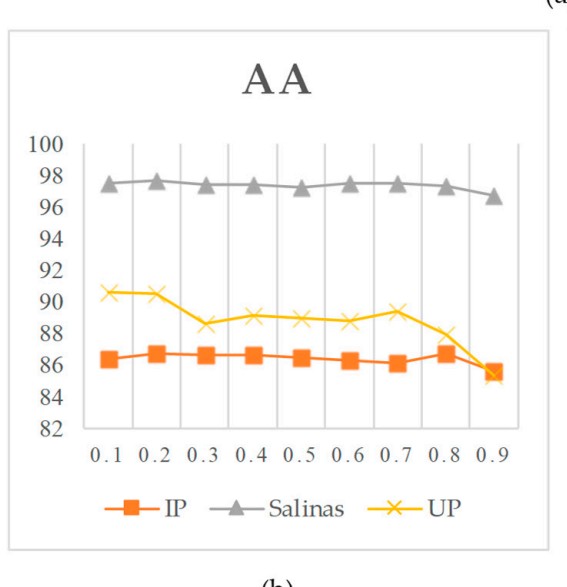

(b)

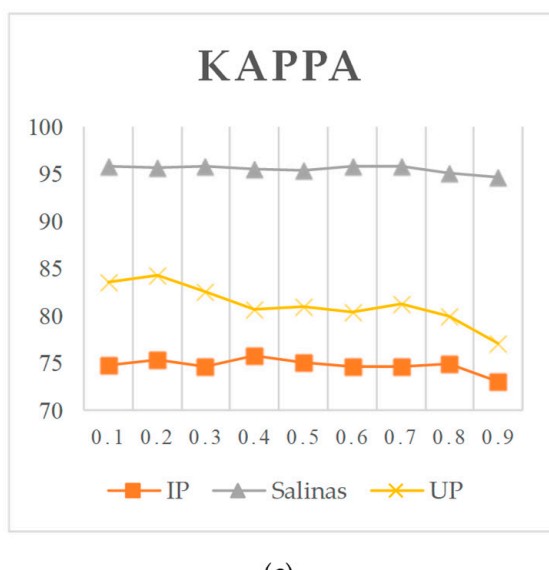

(c)

**Figure 9.** Impact of labeling factors on evaluation indicators under three datasets. (**a**) OA; (**b**) AA; (**c**) Kappa.

**Table 2.** Classification accuracy of Pavia University dataset ablation experiments.

| Number | Category | S3Net | CRSSNet (No SAM) | CRSSNet |
|---|---|---|---|---|
| 1 | Asphalt | **83.72** | 77.46 | 80.05 |
| 2 | Meadows | 82.29 | 87.01 | **89.48** |
| 3 | Gravel | 79.61 | **93.51** | 88.79 |
| 4 | Tress | 91.29 | **92.14** | 91.73 |
| 5 | Sheets | 99.66 | **100.00** | **100.00** |
| 6 | Bare soil | 81.31 | **89.64** | 84.17 |
| 7 | Bitumen | 99.65 | 99.95 | **99.99** |
| 8 | Bricks | 84.73 | 71.67 | **86.29** |
| 9 | Shadow | 91.23 | **94.98** | 93.94 |
| | OA% | 84.40 | 86.19 | **88.00** |
| | AA% | 88.16 | 89.59 | **90.49** |
| | Kappa% | 80.11 | 82.19 | **84.34** |

**Table 3.** Classification accuracy of Indian Pines dataset ablation experiments.

| Number | Category | S3Net | CRSSNet (No SAM) | CRSSNet |
|---|---|---|---|---|
| 1 | Alfalfa | **100.00** | **100.00** | **100.00** |
| 2 | Corn Notill | 59.44 | **76.87** | 75.32 |
| 3 | Corn Mintill | **62.72** | 61.79 | 62.39 |
| 4 | Corn | 92.72 | 99.35 | **99.66** |
| 5 | Grass Pasture | 81.21 | 89.18 | **89.41** |
| 6 | Grass Tree | **92.46** | 91.56 | 90.97 |
| 7 | Grass Pasture Mowed | **100.00** | **100.00** | **100.00** |
| 8 | Hay Windrowed | **99.49** | 98.05 | 97.93 |
| 9 | Oats | **100.00** | **100.00** | **100.00** |
| 10 | Soybean Notill | **73.21** | 60.72 | 58.13 |
| 11 | Soybean Mintill | 65.98 | 63.76 | **68.99** |
| 12 | Soybean Clean | **67.40** | 63.27 | 60.82 |
| 13 | Wheat | **97.35** | 89.85 | 90.30 |
| 14 | Woods | 95.02 | **99.03** | 99.01 |
| 15 | Buildings Grass Tress Drives | 90.21 | 91.99 | **95.33** |
| 16 | Stone Steel Towers | 97.16 | 98.64 | **99.20** |
| | OA% | 75.99 | 77.22 | **78.03** |
| | AA% | 85.89 | 86.50 | **86.72** |
| | Kappa% | 73.08 | 74.47 | **75.33** |

**Table 4.** Classification accuracy of Salinas dataset ablation experiments.

| Number | Category | S3Net | CRSSNet (No SAM) | CRSSNet |
|---|---|---|---|---|
| 1 | Brocoli–Green–Weeds–1 | 99.04 | **100.00** | **100.00** |
| 2 | Brocoli–Green–Weeds–2 | 99.13 | **99.99** | 99.97 |
| 3 | Fallow | 98.80 | **100.00** | **100.00** |
| 4 | Fallow–Rough–Plow | 99.47 | **99.89** | 99.83 |
| 5 | Fallow–Smooth | 95.89 | **96.35** | 95.57 |
| 6 | Stubble | 98.50 | 99.11 | **99.55** |
| 7 | Celery | 99.79 | 99.94 | **100.00** |
| 8 | Grapes–Untrained | 82.63 | 88.16 | **88.91** |
| 9 | Soil–Vinyard–Develop | **99.89** | 99.72 | 99.71 |
| 10 | Corn–Senesced–Green– Weeds | 92.44 | **96.73** | 96.49 |
| 11 | Lettuce–Romain–4wk | **99.50** | 99.29 | 99.45 |
| 12 | Lettuce–Romain–5wk | 93.64 | **98.17** | 97.97 |
| 13 | Lettuce–Romain–6wk | **94.00** | 91.68 | 93.12 |
| 14 | Lettuce–Romain–7wk | 93.05 | **98.42** | 98.21 |
| 15 | Vinyard–Untrained | 93.47 | 92.55 | **94.24** |
| 16 | Vinyard–Vertical–Trellis | **98.82** | 96.81 | 98.06 |
| | OA% | 94.03 | 95.69 | **96.20** |
| | AA% | 96.12 | 97.30 | **97.63** |
| | Kappa × 100% | 93.38 | 95.21 | **95.73** |

From Tables 2–4, it can be observed that there is a significant performance improvement in OA when using the residual volume feature extractor in the network. The sample cases make full use of their hyperspectral data for training. Additionally, after incorporating the spatial-attention mechanism, the network's OA also slightly increases, indicating that the spatial feature extraction part, using SAM after the spatial residual feature extractor extracts, further utilizes spatial information and enhances the network's ability to utilize spatial information, resulting in higher accuracy. Furthermore, from the results of the

Kappa coefficient, it can be observed that CRSSNet also exhibits a significant improvement in terms of consistency compared to CRSSNet (no SAM) and S3Net.

From the ablation experimental-feature classification diagram in Figure 10, it can be observed that the output features are transformed into two-dimensional space using PCA. The Siamese network reduces the intra-class distance and increases the inter-class distance, making the network highly separable. However, compared to S3Net, CRSSNet further improves the separability of the feature maps due to enhanced feature-extraction capabilities, leading to better classification accuracy.

### 4.4. Performance on Cross-Scene HSI Classification

To validate the effectiveness of the proposed method in this paper, classic and advanced hyperspectral image classification methods are selected, including 3DCNN [33], SSRN [34], DFSL + SVM [35], Gia-CFSL [36], DCFSL [37], and S3Net, among other methods. In the aforementioned method experiment, a random selection of five labeled samples per class is performed for training. Comparative analysis is conducted to verify the effectiveness of the proposed method.

Under the setting of a label factor of 0.2, the label-smoothing loss function, the experimental results of the proposed method in this paper and the above-mentioned hyperspectral image classification methods on the Indian Pines, Pavia University, and Salinas datasets are presented in Tables 5–7, respectively.

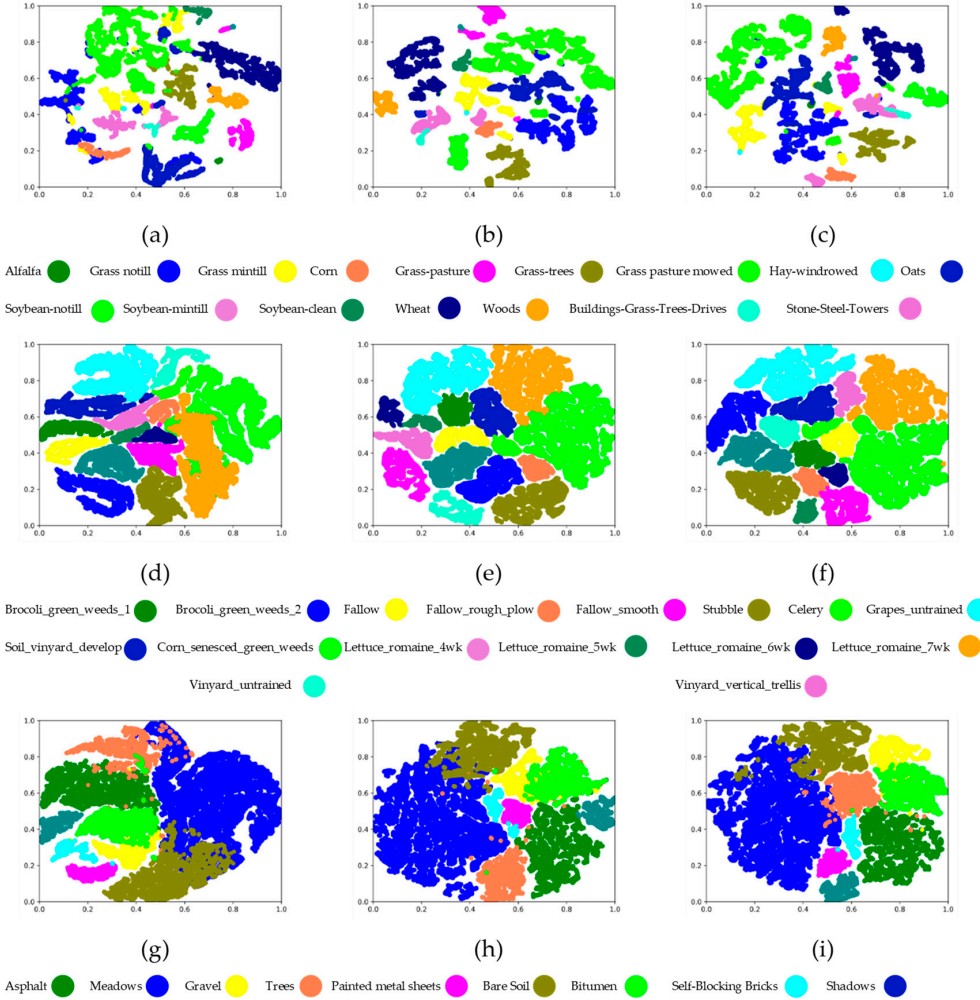

**Figure 10.** Feature separability in ablation experiments. Indian Pines dataset: (**a**) S3Net, (**b**) CRSSNet (no SAM), and (**c**) CRSSNet. Salinas dataset: (**d**) S3Net, (**e**) CRSSNet (no SAM), and (**f**) CRSSNet; Pavia University dataset: (**g**) S3Net, (**h**) CRSSNet (no SAM), and (**i**) CRSSNet.

**Table 5.** Classification accuracy of hyperspectral remote-sensing image dataset of Indian Pines.

| Number | 3DCNN | SSRN | DFSL + SVM | Gia–CFSL | DCFSL | S3Net | Ours |
|--------|-------|------|------------|----------|-------|-------|------|
| 1 | 95.12 | 18.38 | 96.75 | 95.12 | 95.37 | **100** | **100** |
| 2 | 37.70 | 64.79 | 36.38 | 47.36 | 43.26 | 59.44 | **75.32** |
| 3 | 19.77 | 27.65 | 38.34 | 37.94 | 57.95 | **62.72** | 62.39 |
| 4 | 32.51 | 26.97 | 77.16 | 78.45 | 80.60 | 92.72 | **99.66** |
| 5 | 88.45 | 80.76 | 73.92 | 72.80 | 72.91 | 81.21 | **89.41** |
| 6 | 73.65 | 86.87 | 86.25 | 73.38 | 87.96 | **92.46** | 90.97 |
| 7 | 81.82 | 32.24 | 97.10 | **100** | 99.57 | **100** | **100** |
| 8 | 53.35 | **100** | 81.82 | 91.54 | 86.26 | 99.49 | 97.93 |
| 9 | **100** | 57.69 | 75.56 | **100** | 99.33 | **100** | **100** |
| 10 | 41.35 | 59.69 | 52.22 | 66.91 | 62.44 | **73.21** | 58.13 |
| 11 | 66.71 | **70.87** | 59.96 | 67.02 | 62.75 | 65.98 | 68.99 |
| 12 | 37.40 | 45.00 | 36.56 | 27.38 | 48.72 | **67.40** | 60.82 |
| 13 | 85.71 | 88.29 | 98.00 | 96.50 | **99.35** | 97.35 | 90.30 |
| 14 | 62.57 | 97.18 | 84.63 | 91.59 | 85.40 | 95.02 | **99.01** |
| 15 | 56.42 | 36.64 | 74.10 | 63.78 | 66.69 | 90.21 | **95.33** |
| 16 | 90.36 | 60.98 | **100** | 98.86 | 97.61 | 97.16 | 99.20 |
| OA% | 54.76 | 61.36 | 61.69 | 65.75 | 66.81 | 75.99 | **78.03** |
| AA% | 63.93 | 59.75 | 73.05 | 75.54 | 77.89 | 85.89 | **86.72** |
| Kappa × 100% | 48.72 | 56.91 | 56.78 | 61.24 | 62.64 | 73.08 | **75.33** |

**Table 6.** Classification accuracy of hyperspectral remote-sensing image dataset of Pavia University.

| Number | 3DCNN | SSRN | DFSL + SVM | Gia-CFSL | DCFSL | S3Net | Ours |
|--------|-------|------|------------|----------|-------|-------|------|
| 1 | 59.82 | **91.84** | 73.43 | 79.94 | 82.20 | 83.72 | 80.05 |
| 2 | 63.05 | **95.13** | 89.25 | 85.88 | 87.74 | 82.29 | 89.48 |
| 3 | 68.91 | 55.23 | 48.09 | 47.66 | 67.46 | 79.61 | **88.79** |
| 4 | 77.31 | 78.02 | 84.72 | **96.08** | 93.16 | 91.29 | 91.73 |
| 5 | 90.77 | 98.34 | 99.65 | 99.93 | 99.49 | 99.66 | **100** |
| 6 | 63.40 | 53.56 | 67.81 | 65.55 | 77.32 | 81.31 | **84.17** |
| 7 | 87.64 | 60.07 | 64.48 | 77.28 | 81.18 | 99.65 | **99.99** |
| 8 | 57.27 | 85.34 | 67.37 | 62.93 | 66.73 | 84.73 | **86.29** |
| 9 | 95.57 | 98.08 | 92.92 | **99.79** | 98.66 | 91.23 | 93.94 |
| OA% | 65.74 | 76.26 | 79.63 | 79.93 | 83.65 | 84.40 | **88.00** |
| AA% | 73.72 | 79.51 | 76.41 | 79.45 | 83.77 | 88.16 | **90.49** |
| Kappa × 100% | 57.37 | 70.56 | 73.05 | 73.81 | 78.70 | 80.11 | **84.34** |

**Table 7.** Classification accuracy of hyperspectral remote-sensing image dataset of Salinas.

| Number | 3DCNN | SSRN | DFSL + SVM | Gia-CFSL | DCFSL | S3Net | Ours |
|--------|-------|------|------------|----------|-------|-------|------|
| 1 | 95.29 | 97.55 | 73.92 | 97.85 | 99.40 | 99.04 | **100** |
| 2 | 97.20 | 98.97 | 96.85 | 99.76 | 99.76 | 99.13 | **99.97** |
| 3 | 91.45 | 92.47 | 96.28 | 99.54 | 91.96 | 98.80 | **100** |
| 4 | 97.31 | 96.50 | 99.11 | 95.39 | 99.55 | 99.47 | **99.83** |
| 5 | 91.24 | 94.20 | 80.72 | 91.58 | 92.70 | **95.89** | 95.57 |
| 6 | 98.80 | 99.28 | 91.63 | **99.80** | 99.52 | 98.50 | 99.55 |
| 7 | 99.69 | 99.98 | 97.73 | 98.77 | 98.88 | 99.79 | **100** |
| 8 | 66.40 | 86.90 | 82.33 | 77.36 | 74.57 | 82.63 | **88.91** |

**Table 7.** *Cont.*

| Number | 3DCNN | SSRN | DFSL + SVM | Gia-CFSL | DCFSL | S3Net | Ours |
|---|---|---|---|---|---|---|---|
| 9 | 96.25 | 99.64 | 94.44 | 99.14 | 99.59 | **99.89** | 99.71 |
| 10 | 70.72 | 92.01 | 80.96 | 66.97 | 96.42 | 92.44 | **96.49** |
| 11 | 93.15 | 95.86 | 93.38 | 90.59 | 96.61 | **99.50** | 99.45 |
| 12 | 99.65 | 99.15 | 97.94 | **99.95** | 99.93 | 93.64 | 97.97 |
| 13 | 92.63 | 83.24 | 95.79 | **99.34** | 99.30 | 94.00 | 93.12 |
| 14 | 93.56 | 95.15 | **98.87** | 98.40 | 98.85 | 93.05 | 98.21 |
| 15 | 68.02 | 55.97 | 91.13 | 70.45 | 75.38 | 93.47 | **94.24** |
| 16 | 81.41 | **98.91** | 90.57 | 92.97 | 92.22 | 98.82 | 98.06 |
| OA% | 84.20 | 86.39 | 86.95 | 88.00 | 89.34 | 94.03 | **96.20** |
| AA% | 89.56 | 93.24 | 90.08 | 92.36 | 94.04 | 96.12 | **97.63** |
| Kappa × 100% | 82.46 | 84.95 | 85.51 | 86.64 | 88.17 | 93.38 | **95.73** |

According to the results of Indian Pines, Pavia University, and Salinas datasets in Table 5, Table 6, and Table 7, respectively, this method achieves the highest overall classification accuracy of 78.03%, 88.00%, and 96.20%. Compared with Siamese network S3Net in transfer learning, the overall classification accuracy of the proposed algorithm increased by 2.04%, 3.6%, and 2.17%, the average classification accuracy increased by 0.83%, 2.33% and 1.51%, and the Kappa coefficient increased by 0.0225, 0.0423, and 0.0235, respectively. Compared with cross-domain method DCFSL in transfer learning, the overall classification accuracy increased by 11.22%, 4.35%, and 6.86%, the average classification accuracy increased by 8.83%, 6.72%, and 3.59%, and the Kappa coefficient increased by 0.1269, 0.0564, and 0.0756, respectively. This method improves the network structure by using a residual feature extractor and introducing a spatial-attention mechanism. These enhancements enable the feature-extraction network to have better feature extraction capabilities for both spectral and spatial information in hyperspectral data. Additionally, the inclusion of the label smooth cross-entropy loss function helps reduce network overfitting. Compared to alternative approaches, the classification performance of hyperspectral images can be notably enhanced, especially in scenarios with limited sample sizes. Experimental results from three distinct datasets demonstrate the superior consistency of the proposed method.

As depicted in the visualization results presented in Figures 11–13, CRSSNet exhibits superior classification performance compared to other methods. In the Indian Pines dataset shown in Figure 11, it is evident that different classification methods exhibit certain classification errors and fail to achieve satisfactory results with limited samples. However, when compared to other methods, the proposed method achieves the best classification performance in the categories of Corn and Buildings Grass Tress Drives, among others. In the classification results of the Pavia University dataset in Figure 12, compared to other methods, CRSSNet shows lower misclassifications in the categories of Sheets and Bare soil, resulting in a superior classification effect map. In the Salinas dataset, shown in Figure 13, the classification effect map generated by CRSSNet closely resembles the real image, with only a few misclassifications observed in individual categories such as Grapes untrained. However, in categories like Brocoli Green Weeds_2 and Fallow, among others, CRSSNet demonstratess the best classification performance.

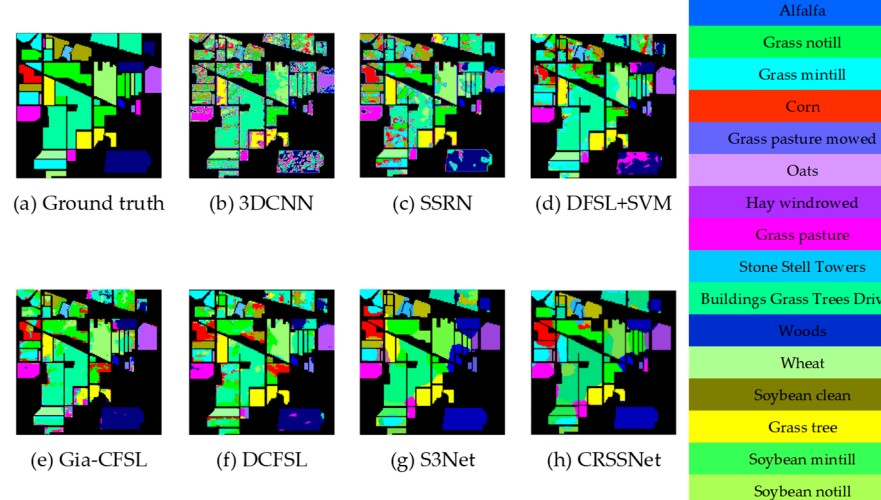

**Figure 11.** Results of classification for Indian Pines dataset.

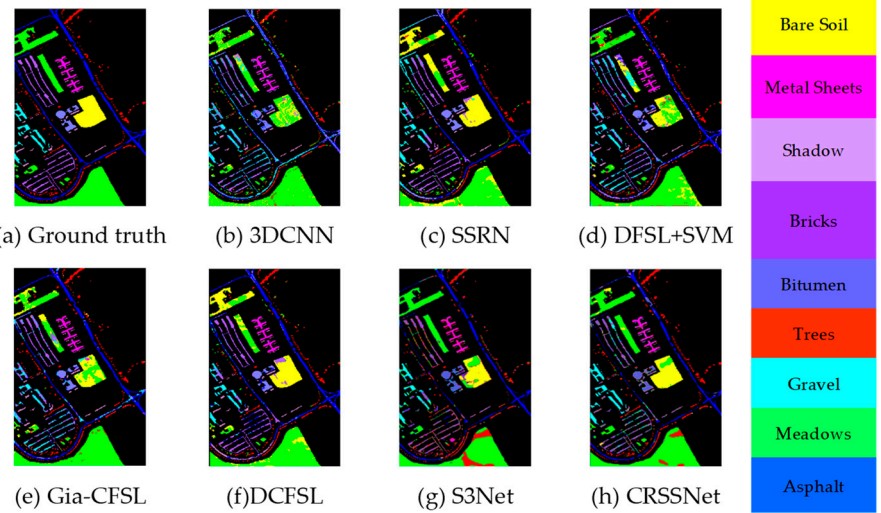

**Figure 12.** Results of classification for Pavia University dataset.

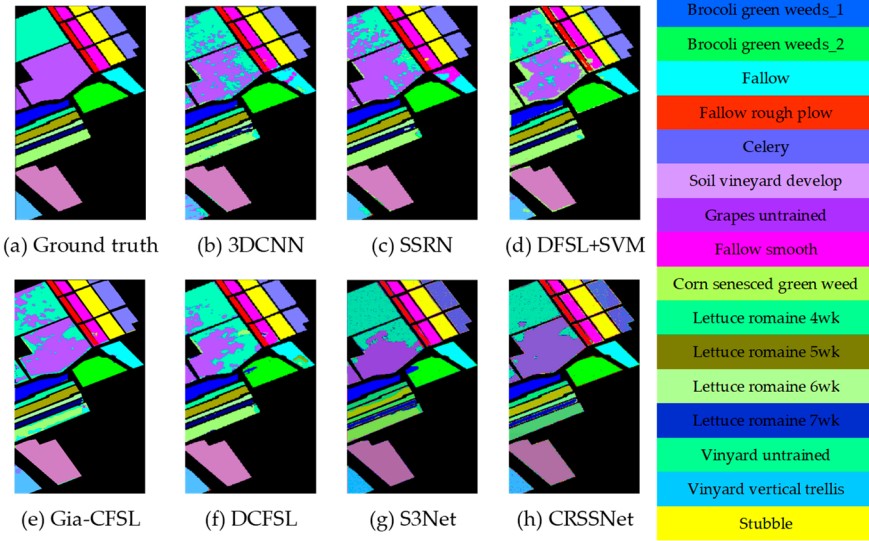

**Figure 13.** Results of classification for Salinas dataset.

## 5. Conclusions

To address the challenges of insufficient spatial and spectral information extraction, as well as network overfitting in small-sample hyperspectral image classification, a method based on convolutional residuals and SAM Siamese networks is proposed. This method incorporates a feature residual-extraction module and a spatial attention mechanism to enhance feature information extraction. Furthermore, a more effective loss function is employed, which introduces noise and explores label factors to improve the classification capability of hyperspectral images with limited samples. Experimental results demonstrate the effectiveness of the proposed method on publicly available hyperspectral datasets, including Indian Pines, Pavia University, and Salinas. When compared to other methods, the proposed approach achieves superior classification performance in metrics such as OA, AA, and Kappa. These results highlight the method's enhanced classification performance and improved generalization ability. However, there are still some challenges that need to be addressed. As observed from Figures 11–13, the classification accuracy is relatively low when dealing with similar categories.

Currently, there are still challenges in solving the classification of hyperspectral images with limited samples, especially regarding the inherent high-dimensional nature of such images. Although our proposed network incorporates PCA for dimensionality reduction, the issue of high dimensionality persists due to the oversampling of the spectral dimensions in hyperspectral images. Recently, some graph-based methods for dimensionality reduction of hyperspectral images have been proposed. For instance, Luo et al. [38] introduced an Enhanced Hybrid Graph Discriminative Learning (EHGDL) based on hypergraphs, while Zhang et al. [39] presented Multilayer Graph Spectral Analysis of Hyperspectral Images using Multilayer Graph Signal Processing (M–GSP), among others, which have shown promising results. In future research, it would be beneficial to explore and build upon these graph-based methods to achieve better dimensionality reduction performance in hyperspectral image classification tasks.

Moreover, despite the need for a limited number of labeled samples in hyperspectral image classification, the reliance on manual labeling for training remains crucial. Therefore, future research should explore the potential of conducting hyperspectral image classification without the dependence on labeled samples. Such developments hold significant promise in contributing to the field and its practical applications.

**Author Contributions:** Conceptualization, M.X.; methodology, L.Y. and K.X.; software, M.X.; writing—original draft preparation, M.X.; writing—review and editing, H.Z. and G.Y.; supervision, Y.R. and Z.S. All authors have read and agreed to the published version of the manuscript.

**Funding:** This research was funded by Major Science and Technology Project in Yunnan Province, grant number 202202AD080004. The APC was funded by Yunnan Province Science and Technology Department.

**Data Availability Statement:** The datasets presented in this work are openly available on the website. Available online: https://www.ehu.eus/ccwintco/index.php/Hyperspectral_Remote_Sensing_Scenes (accessed on 6 August 2023).

**Conflicts of Interest:** The authors declare no conflict of interest.

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
