# Peer review of "Few-Shot Hyperspectral Image Classification Based on Convolutional Residuals and SAM Siamese Networks"

_electronics, doi:10.3390/electronics12163415_

Round 1

Reviewer 1 Report

This work is well presented. The methodology and network structure are well described. Adequate experiments have been performed to validate the proposed network. However, the experiment results do not show the superiority of CRSSNet, especially in Table 4. It would be much nicer if more explanations were provided. I am also very curious if the performance would be improved by switching the SAM model with a transformer. Overall it is well-written scientific work with moderate novelty. The results well support the conclusion. 

Author Response

I sincerely appreciate your comprehensive evaluation of our research work and the valuable insights you provided. As you rightly pointed out, the performance of neural network models can vary across different datasets, with some datasets showing promising results while others may be less optimistic. This indeed remains one of the challenges in current research on hyperspectral image classification.

We have acknowledged this issue in the conclusion section of our paper and have made some modifications to the experimental analysis accordingly. Furthermore, we carefully consider your suggestion of exploring the replacement of the "SAM model with a Transformer" as a potential research direction. We plan to conduct experimental validations in our future studies to investigate the effectiveness of the proposed approach.

Once again, thank you for your valuable feedback on our paper. Your constructive criticism will undoubtedly help us improve our research and contribute to the advancement of hyperspectral image classification.

Author Response

Response 1: We sincerely appreciate your review of our paper and the valuable suggestions provided. Following your advice, we have incorporated brief introductions to each section in the introduction, facilitating readers' comprehension.

Response 2: In response to your revision suggestions, we have conducted thorough discussions and made careful amendments, updating and expanding some of the international literature related to our research content from 2022 and 2023.

Response 3: We extend our sincere apologies for any confusion arising from the imprecise terminology employed in our previous discourse. Our original intent was to elucidate that the Kappa coefficient was employed to demonstrate the robust consistency of our network. Specifically, within the ambit of this experimental investigation, the Kappa coefficient exhibited favorable responses compared to other networks, substantiating our chosen research methodology. Regrettably, due to our prior inadequacies in expression, the context of our paper may have been misconstrued.

In response to this, we have effectuated requisite modifications within the manuscript to aptly and accurately convey our intended message. We wish to convey our heartfelt gratitude for your diligent review and invaluable feedback. Your understanding is greatly appreciated.

Reviewer 3 Report

The paper is difficult to understand due to the assumption of a solid initial context. It is taken for granted that the readers are familiar with the project's background. However, I find the topic quite interesting, My recommendations would be to provide more context in the introduction and to review or provide functional links in the cited references, as some of them are inactive. 

The written English appears to be comprehensible 

Author Response

Response 1: First and foremost, we wish to express our gratitude for your meticulous review of our paper and the invaluable suggestions you have provided. In alignment with your recommendations, we have augmented the introduction with pertinent literature relevant to our study, incorporating online links for readers to access each cited reference. Moreover, introductory remarks have been appended to each section within the introduction, facilitating enhanced reader comprehension of the paper's organizational structure. Lastly, within the conclusion section, we have undertaken an analysis of the prevailing challenges within the domain of small-sample hyperspectral image classification research, while also delineating prospective directions for future investigations. Once again, we extend our heartfelt appreciation for your thorough peer review and insightful feedback.

Reviewer 4 Report

The authors proposed a novel framework for a few-shot hyperspectral image classification problem. The framework is reasonable. My major concerns are for the experimental results. Here are my comments.

1. In the comparison, most of the benchmarks are before 2020. Please compare with more state-of-the-art approaches, such as some 2022 and 2023 works, to demonstrate the effectiveness of the proposed methods.

2. How is the robustness of the proposed method against wrong labeling and noise? Can the authors provide some results with noise and wrong labels since a few-shot learning setup shall be more efficient to address these scenarios than traditional learning frameworks?

3. Please fix typos. For example, in line 300 page 9, it shall be "In equation..." Please also check other parts to fix typos.

4. Please also discuss the limitations of the few-shot learning in HSI classification and suggest several future directions after the conclusion.

See the major comments.

Author Response

Response 1: We extend our sincere gratitude for your diligent review of our paper and the invaluable suggestions provided. In accordance with your recommendations, we have expunged the antiquated SVM network and introduced the 2022 graph convolution-based few-shot learning hyperspectral image classification network (Gia-CFSL) as a comparative methodology within our experimental comparisons. Presently, our comparative experiments encompass three classical classification approaches (3DCNN, SSRN, and DFSL+SVM), alongside three novel classification models introduced in 2022 (Gia-CFSL, DCFSL, and S3Net). Through juxtaposition with the aforementioned research methodologies, we hold the conviction that the effectiveness of our proposed approach can be robustly substantiated.

Response 2: We deeply regret any confusion arising from imprecise terminology employed in our manuscript. Our original intention was to convey that the Kappa coefficient was utilized to signify the network's favorable consistency. Presently, the evaluation of hyperspectral image classification predominantly relies upon three metrics: Overall Accuracy (OA), Average Accuracy (AA), and the Kappa coefficient, thereby rendering our proposed method's efficacy contingent upon the outcomes of these three metrics. As our paper does not delve into a discussion of robustness, we unfortunately lack the relevant content to address the robustness against noise and erroneous labels. We sincerely apologize for this limitation. At present, we have undertaken the necessary amendments to accurately and appropriately convey our message. Finally, we extend our heartfelt appreciation for your diligent review and invaluable feedback, particularly your understanding regarding our apology.

Response 3: We express our gratitude for your diligent review of our manuscript and for highlighting areas of improvement. We have duly addressed the identified issues and made the necessary modifications in accordance with your suggestions. Your invaluable feedback has significantly contributed to the enhancement of our work.

Response 4: We appreciate your suggestions regarding our conclusion. Based on the original conclusion, we have included additional discussions on current research challenges, such as the high dimensionality of hyperspectral images and variations in classification performance across different datasets. Furthermore, we have addressed future prospects, such as exploring the issue of unknown classes in the context of few-shot scenarios.

Reviewer 5 Report

Critical response on a manuscript entitled “Few-Shot Hyperspectral Image Classification Based On Convolutional Residuals And SAM Siamese Networks” presented by Mengen Xia et al.

The submitted paper is the work of deserve. It’s main goal is development of hyperspectral image classification engine based on proposed CRSSNet model. As for profit of new model engineering authors stated better classification results shown for testing datasets. Group of authors effectively described existing classification methods applicable for hyperspectral data, their flaws and advantages. Also, their well presented comparison of machine learning, deep learning and few-shot learning in Introduction. The “related work” section gives exemptive outline of traditional methods and works of predecessors and presents proposed methodology as based on previous studies. 

Special “Introduction to the methodology” section shows network architecture and describes its constituent parts in a thorough and comprehensive manner. 

Results shown in Section 4 and especially accuracy tests are impressive. 

However, below we will present a few less significant issues, which resolving could make manuscript more valuable for the international readers. 

Line 67. ‘Metric learning’ and ‘meta-learning’ are not exactly mentioned on Figure 1(c), which is ambiguous. 

Figure 2; 3. Text size is to small to be read. 

Line 310. Reference on datasets should be placed and mentioned in a reference lists. 

Line 324. Pytorch library deserves reference. 

Line 471. Link on datasets IS NOT WORKING. 

Reproducibility of results requires sources codes to be available, it is hard to reproduce authors results even knowing Pytorch, cause model adjustment is important. I recommend to follow open source way and make source codes available. 

I also propose to change structure of work, rename “Introduction to the methodology” section into “Materials and methods” and add description of used datasets. 

Conclusion. The paper under consideration is of good quality and well deserved. I recommend publication after minor possible amendments.

Author Response

Response 1: We would like to express our sincere appreciation for your diligent review of our paper and valuable suggestions. As per your advice, we have incorporated the terms "metric learning" and "meta-learning" in Figure 1(c). Additionally, we have made the necessary adjustments to the text in Figure 2 and Figure 3.

Response 2: We extend our gratitude for your insightful recommendations regarding the datasets. We have duly incorporated the dataset links within the paper and included indices for quick accessibility in the dataset descriptions. Given that the subsequent section, "Experimental Datasets," encompasses both concise dataset introductions and comprehensive information tables, the provision of additional elaborate dataset descriptions may indeed be deemed superfluous. We appreciate your attention to these suggestions and the endeavors undertaken to enhance the manuscript. Should you require any further inquiries or assistance, please do not hesitate to reach out.

Response 3: Thank you for your suggestions. Our research methodology is conducted within the PyTorch framework. As our forthcoming phase of study builds upon the foundation of this current research approach, we regretfully find ourselves in a position where we are unable to release the source code at present. We sincerely appreciate your understanding in this matter.

Round 2

Reviewer 4 Report

The authors have addressed most of my concerns. Please proofread the manuscript to fix typos. Then it shall be ready.

Please also consider including the following graph-based references when discussing dimension reduction in future direction:

[1]  Luo, F., Zhang, L., Du, B., & Zhang, L. (2020). Dimensionality reduction with enhanced hybrid-graph discriminant learning for hyperspectral image classification. IEEE Transactions on Geoscience and Remote Sensing, 58(8), 5336-5353.

[2] Zhang, S., Deng, Q., & Ding, Z. (2022). Multilayer graph spectral analysis for hyperspectral images. EURASIP Journal on Advances in Signal Processing, 2022(1), 92.  

Author Response

We are honored to receive your response once again. We greatly appreciate your diligent review and invaluable suggestions. Based on your suggestions, we have added a graph-based dimensionality reduction method to the conclusion section and cited the two references you provided. Finally, thank you for affirming our article.
